# The Origin of Stroma Influences the Biological Characteristics of Oral Squamous Cell Carcinoma

**DOI:** 10.3390/cancers13143491

**Published:** 2021-07-12

**Authors:** Haruka Omori, Qiusheng Shan, Kiyofumi Takabatake, Keisuke Nakano, Hotaka Kawai, Shintaro Sukegawa, Hidetsugu Tsujigiwa, Hitoshi Nagatsuka

**Affiliations:** 1Department of Oral Pathology and Medicine, Okayama University Graduate School of Medicine, Dentistry and Pharmaceutical Science, Okayama 700-8525, Japan; p4628fuz@s.okayama-u.ac.jp (H.O.); p0h53x0n@s.okayama-u.ac.jp (Q.S.); pir19btp@okayama-u.ac.jp (K.N.); de18018@s.okayama-u.ac.jp (H.K.); gouwan19@gmail.com (S.S.); tsuji@dls.ous.ac.jp (H.T.); jin@okayama-u.ac.jp (H.N.); 2Department of Oral and Maxillofacial Surgery, Kagawa Prefectural Central Hospital, Kagawa 760-0065, Japan; 3Department of Life Science, Faculty of Science, Okayama University of Science, Okayama 700-0005, Japan

**Keywords:** gingival ligament tissue-derived stromal cells, periodontal ligament tissue-derived stromal cells, oral squamous cell carcinoma, tumor microenvironment, biological character

## Abstract

**Simple Summary:**

Normal stromal cells play a significant role in the progression of cancers but are poorly investigated in oral squamous cell carcinoma (OSCC). In this study, we found that stromal cells derived from the gingival and periodontal ligament tissues could inhibit differentiation and promote the proliferation, invasion, and migration of OSCC both in vitro and in vivo. Furthermore, microarray data suggested that genes, such as *CDK1*, *BUB1B*, *TOP2A*, *DLGAP5*, *BUB1*, and *CCNB2*, probably play a role in influencing the different effects of gingival stromal tissue cells (G-SCs) and periodontal ligament stromal cells (P-SCs) on the progression of OSCC. Therefore, both G-SCs and P-SCs could promote the progression of OSCC, which could be a potential regulatory mechanism in the progression of OSCC.

**Abstract:**

Normal stromal cells surrounding the tumor parenchyma, such as the extracellular matrix (ECM), normal fibroblasts, mesenchymal stromal cells, and osteoblasts, play a significant role in the progression of cancers. However, the role of gingival and periodontal ligament tissue-derived stromal cells in OSCC progression is unclear. In this study, the effect of G-SCs and P-SCs on the differentiation, proliferation, invasion, and migration of OSCC cells in vitro was examined by Giemsa staining, Immunofluorescence (IF), (3-(4,5-dimethylthiazol-2-yl)-5-(3-carboxymethoxyphenyl)-2-(4-sulfophenyl)-2H-tetrazolium) (MTS), invasion, and migration assays. Furthermore, the effect of G-SCs and P-SCs on the differentiation, proliferation, and bone invasion by OSCC cells in vivo was examined by hematoxylin-eosin (HE) staining, immunohistochemistry (IHC), and tartrate-resistant acid phosphatase (TRAP) staining, respectively. Finally, microarray data and bioinformatics analyses identified potential genes that caused the different effects of G-SCs and P-SCs on OSCC progression. The results showed that both G-SCs and P-SCs inhibited the differentiation and promoted the proliferation, invasion, and migration of OSCC in vitro and in vivo. In addition, genes, including *CDK1*, *BUB1B*, *TOP2A*, *DLGAP5*, *BUB1*, and *CCNB2*, are probably involved in causing the different effects of G-SCs and P-SCs on OSCC progression. Therefore, as a potential regulatory mechanism, both G-SCs and P-SCs can promote OSCC progression.

## 1. Introduction

Cancer proliferation and progression occur in concert with alterations in the surrounding stroma. Indeed, the tumor microenvironment (TME) is now recognized as a critical element for tumor development and progression. Many studies have suggested that in solid tumors the tumor parenchyma and stroma influence each other [1]. Several recent studies have shown that in some tumors, such as pancreatic tumors, tumor cells transplanted with the stroma alter the ability of the tumor to spread and influence its prognosis [2,3,4,5]. However, it remains unclear how the tumor stroma is directly involved in the growth, infiltration, and metastasis of the tumor parenchyma.

Oral squamous cell carcinoma (OSCC) is the most common malignant neoplasm of the oral cavity [6,7]. OSCC is a malignant epithelial tumor; however, it is similar to many other solid tumors, wherein their infiltration and proliferation affect the interaction between the tumor parenchyma and stroma [8]. OSCC occurs most frequently on the tongue followed by the gingiva [9,10]. OSCC derived from the tongue epithelium invades the sublingual tissue. However, only connective tissue and muscle tissue are present in the vertical and horizontal directions of OSCC infiltration, and the environment in which the cancer cells are in contact does not change regardless of the degree of cancer cell invasion. On the other hand, OSCC developing in the gingival epithelium invades the subgingival connective tissue, destroys the jawbone in vertical infiltration, or reaches the periodontal ligament tissue in horizontal infiltration. The TME of OSCC derived from the gingival epithelium is complex, and its TME changes depending on its infiltration direction. Based on the above, the environment surrounding the tumor parenchyma promotes cancer growth and infiltration changes dramatically in OSCCs occurring in the gingival epithelium compared to OSCCs developing in the tongue epithelium.

In our previous study, we isolated human primary tumor stromal cells from two patients with OSCC having a different invasion ability, and their stromal cells were co-cultured with the same cancer cells line. This study demonstrated that the stroma altered the proliferation and invasion of OSCC cells in vitro and changed the biological characteristics of the tumor in vivo, despite using the same cancer cells line [11]. These results suggested that the tumor stroma may directly control the biological character of the tumor parenchyma. Cancer stromal cells derived from normal tissues are formed by educating the tumor parenchyma [12,13]. OSCC developing in the gingival epithelium induces the tumor stroma by educating normal tissues with different origins and constituents, such as gingival connective tissue or periodontal ligament. Therefore, we hypothesized that the tumor stroma derived from gingival connective tissue or the periodontal ligament might have different biological characteristics and might provide different effects of the tumor parenchyma even if the same tumor parenchyma is in contact with the tumor stroma of different origin.

In this study, we aimed to investigate how tumor tissue surrounding normal tissues, such as gingival connective tissue and the periodontal ligament, affects the biological characteristics of the tumor parenchyma. Towards this, gingival connective tissue and the periodontal ligament isolated from human tissues, a well-differentiated squamous cell carcinoma cell line (HSC-2) and a poorly differentiated squamous cell carcinoma cell line (HSC-3) were used to examine the degree of tumor differentiation, proliferative capacity, and bone infiltration.

## 2. Materials and Methods

### 2.1. Isolation of Gingival Stromal Cells and Periodontal Ligament Stromal Cells

Primary cultured gingival stromal tissue cells (G-SCs) and periodontal ligament stromal cells (P-SCs) from human tissues were isolated. The gingival tissue was collected from the gingiva attached around the extracted crown to prevent pericoronitis. In addition, the periodontal ligaments from a single human tooth were dissected free of the root surface. The gingival tissue and periodontal ligaments were collected from the same one patient and confirmed to be free of lesions by radiology and oral surgery. These cell samples were obtained from tissues post-surgery at the Oral Surgery Department of Okayama University. This study was approved by the Ethics Committee of Okayama University (project identification code: 1703-042-001, approval date: 10 March 2017). Informed consent was obtained from all patients. Pieces of fresh tissue (1 mm^3^) were washed several times with alpha-Modified Eagle Medium (MEM) (Life Technologies, Thermo Fisher Scientific Inc. Waltham, MA, USA) containing antibiotic–antimycotic solution (Life Technologies, Thermo Fisher Scientific Inc. MA, USA) and then minced. These tissues were treated with alpha-MEM containing 1 mg/mL collagenase II (Invitrogen Co., New York, NY, USA) and dispase (Invitrogen Co., New York, NY, USA) for 2 h at 37 °C with shaking (200 rpm). The released cells were centrifuged for 5 min at 1000 rpm, suspended in alpha-MEM containing 10% FBS (Biowest, Nuaillé, France), filtered using a cell strainer (100 μm, BD Falcon, BD Bioscience, Primus, UK), plated in a tissue culture flask, and incubated at 37 °C in 5% CO_2_. One week later, the cells were treated with Accutase (Invitrogen Co.) based on the different adhesion properties of epithelial and gingival connective tissue cells/periodontal ligament cells. In this experiment, human dermal fibroblasts (HDFs) purchased from LONZA (Tokyo, Japan) were used as a control since human dermal fibroblasts are not of oral origin and present close similarity to squamous cell carcinoma. G-SCs, P-SCs, and HDFs were maintained in alpha-MEM containing 10% FBS and were used within 10 passages to eliminate transformation due to passaging.

### 2.2. OSCC Cell Lines

The well-differentiated human OSCC cell line HSC-2 and the poorly differentiated human OSCC cell line HSC-3 were purchased from the Cell Bank of the Japanese Collection of Research Bioresources (JCRB, Osaka, Japan) and maintained in alpha-MEM supplemented with 10% FBS and 100 U/mL antimycotic-antibiotic solution at 37 °C in a humidified atmosphere with 5% CO_2_.

### 2.3. Tumor Morphology

The stromal cells (G-SCs, P-SCs, and HDFs) and cancer cells (HSC-2 and HSC-3) were mixed at a ratio of 3:1, respectively. The mixed cells were seeded in a small dish with a slide at a density of 4 × 10^5^ cells/dish. After incubation for 1 d, the slides, with the cells attached, were stained with the Giemsa Staining Kit (Diff-Quick, Nanjing Jiancheng Bioengineering Institute, Nanjing, China). The stained cells were photographed using a bright field microscope (BX51, Olympus, Tokyo, Japan), and the experiment was repeated three times, independently.

### 2.4. Tumor Proliferation Assay in 2D Co-Culture (MTS Assay)

The stromal cells (G-SCs, P-SCs, and HDFs) and cancer cells (HSC-2 and HSC-3) were lysed and centrifuged when the density of cells approached nearly 90%. Then, G-SCs, P-SCs, HDFs, and HSC-2, -3 cells were mixed at a ratio of 3:1. The mixed cells were seeded into 96-well plates at a density of 2000 cells/well. After incubation for 1, 3, and 5 days, 20 μL of MTS reagent (CellTiter 96 Aqueous One Solution Cell Proliferation Assay, Promega Corporation, WI, USA) were added into each well and incubated for 4 h. The absorbance of each well was measured at 490 nm using an enzyme-linked immunosorbent assay reader. The experiment was repeated three times, independently, and the data were analyzed using GraphPad Prism 9.

### 2.5. Tumor Proliferation Assay in 2D Co-Culture (Ki-67 Labeling Index)

Proliferation measurements in 2D co-culture were performed using Ki-67, a proliferation-specific marker. The stromal cells (G-SCs, P-SCs, and HDFs) and cancer cells (HSC-2 and HSC-3) were mixed at a ratio of 3:1, respectively. The mixed cells were seeded in a small dish with a slide at a density of 4 × 10^5^ cells/dish and cultured for two days. The slides were washed three times with tris-buffered saline (TBS), and the cells were then fixed with 4% paraformaldehyde for 15 min. Endogenous peroxidase activity was blocked by incubating the slides in 0.3% H_2_O_2_ in methanol for 30 min, and the cells were blocked with blocking solution for 20 min. The mouse anti-Ki-67 (1:50, DAKO, Tokyo, Japan) primary antibody was added and incubated for 2 h. After washing three times with TBS, all the slides were incubated with secondary antibodies (avidin-biotin complex-based detection) (rabbit/goat/mouse ABC kit; Vector Laboratories, Inc., Burlingame, CA, USA) for 1 h at room temperature. Then, the slides were visualized using a diaminobenzidine (DAB)/H_2_O_2_ mixed solution (Histofine DAB substrate; Nichirei, Tokyo, Japan), and the section was counterstained with Mayer’s hematoxylin. The experiment was repeated three times, independently.

### 2.6. Invasion Assay and Migration Assay

The stromal cells (G-SCs, P-SCs, and HDFs) and cancer cells (HSC-2 and HSC-3) were lysed and centrifuged when the cell density approached nearly 90%. Next, normal stromal cells and cancer cells were mixed at a ratio of 3:1 in alpha-MEM without FBS. For the invasion assay, cells were seeded into the upper chamber on 8 μm pore size transwell filters that were precoated with Matrigel in 24-well plates (Corning BioCoat Matrigel Invasion Chamber Kit, BD Biosciences, San Jose, CA, USA) at a density of 4 × 10^4^ cells/500 μL. Alpha-MEM (500 μL) with 10% FBS was added to the lower chamber. After incubation for 1 d, the cells in the upper chamber were removed with a cotton swab, and the membrane was cut in the upper chamber. For the migration assay, cells were seeded into the upper chamber on 8 μm pore size transwell filters without Matrigel in 24-well plates at a density of 2 × 10^4^ cells/500 μL. Alpha-MEM (500 μL) with 10% FBS was added to the lower chamber. After incubation for 1 d, the cells in the upper chamber were removed with a cotton swab, and the membrane was cut in the upper chamber.

### 2.7. Double-Fluorescence Immunohistochemical Staining

To determine whether the cells that invaded or migrated were cancer cells or normal stromal cells, we performed double-fluorescence immunohistochemical staining. After washing the invasion or migration membranes three times with TBS (5 min each time), the cells were fixed with 4% paraformaldehyde for 15 min and blocked with blocking solution for 20 min. The primary rabbit anti-vimentin (1:200, Abcam, MA, USA) and mouse anti-AE 1/3 (Abcam) antibodies were added and incubated for 1 h. After washing three times with TBS, the anti-mouse IgG Alexa Fluor 488 (1:200, Life Technologies, Carlsbad, CA, USA) and anti-rabbit IgG Alexa Fluor 568 (1:200, Life Technologies) secondary antibodies were added and incubated for 1 h without light. After washing three times with TBS and distilled water (DW) for three times, respectively, the samples were stained with 0.2 g/mL 4,6-diamidino-2-phenylindole (DAPI) (Dojindo Laboratories, Kumamoto, Japan). The staining results were observed using the All-in-One, BZ-X700 fluorescence microscope (Keyence, Osaka, Japan). The experiment was repeated three times, independently.

### 2.8. Experimental Animals

All animal experiments were conducted according to the relevant guidelines and regulations approved by the institutional committees at Okayama University (OKU-2017406). After being anesthetized intraperitoneally with ketamine hydrochloride (75 mg/kg body weight) and medetomidine hydrochloride (0.5 mg/kg body weight), the mixed cell groups, including HSC-2, -3 cells (1 × 10^6^) and normal cells (G-SCs, P-SCs, HDFs, 3 × 10^6^) were injected into the central region of the top of the head (lamina propria) of BALB/c nu-nu mice (4-week-old healthy females). Finally, atipamezole hydrochloride (1 mg/kg body weight) was injected subcutaneously to induce awakening.

### 2.9. Hematoxylin-Eosin Staining (HE Staining)

After 4 weeks, the tumor tissues and surrounding bone tissues were removed, fixed with 4% paraformaldehyde for 12 h, and decalcified with 10% EDTA for 4 weeks. Subsequently, the tissues were processed and embedded into paraffin wax by routine histological preparation and further cut into 5 μm sections. Finally, the sections were stained with hematoxylin and eosin.

### 2.10. Cell Proliferation Assay (Ki-67 Labeling Index)

For in vivo tissue xenografts, proliferation measurements in paraffin-embedded samples of tumor explants were performed using Ki-67. Endogenous peroxidase activity was blocked by incubating the sections with 0.3% H_2_O_2_ in methanol for 30 min. Antigen retrieval was achieved by heat treatment using a 10 mM citrate buffer solution (pH 6.0). After antigen retrieval, the sections were blocked with 10% normal serum for 20 min at room temperature and incubated with mouse anti-Ki-67 primary antibody (1:50, DAKO) overnight at 4 °C. After washing three times with TBS, all the sections were incubated with the secondary antibodies (avidin-biotin complex-based detection) (Rabbit/Goat/mouse ABC kit; Vector Laboratories, Inc., Burlingame, CA, USA) for 1 h at room temperature. The sections were then visualized using a diaminobenzidine (DAB)/H_2_O_2_ mixed solution (Histofine DAB substrate; Nichirei, Tokyo, Japan). The experiment was repeated three times, independently.

### 2.11. Tartrate-Resistant Acid Phosphatase Staining (TRAP Staining)

TRAP staining was performed using the TRAP staining kit (Primary Cell, Sapporo, Japan) according to the manufacturer’s instructions. The activated multi-nucleated osteoclast cells on the bone resorption surface were counted using Image J software (NIH, Bethesda, MD, USA).

### 2.12. Microarray and Bioinformatics Analyses

The differentially expressed genes (DEGs) between G-SCs and P-SCs were analyzed by microarray, and |Log_2_FC| > 1 was considered as the cutoff value. In summary, RNA was prepared using the miRNeasy micro kit following the manufacturer’s recommendations. In addition, the concentration of RNA was tested by Nanodrop One (Thermo Fisher Scientific). Cyanine-3 (Cy3)-labeled cRNA was prepared from 100ng RNA. Dye incorporation and cRNA yield were checked with the BioanalyzerRNA6000 Nano. A total of 100 ng of Cy3-labelled cRNA was fragmented at 60 °C for 30 min in a reaction volume of 250 mL containing a 1× Agilent fragmentation buffer and a 2× Agilent blocking agent following the manufacturer’s instructions. On completion of the fragmentation reaction, 250 mL of a 2× Agilent hybridization buffer was added to the fragmentation mixture and hybridized to the Agilent SureScan Microarray Scanner G4900DA (Agilent Technologies, Agilent, CA, USA) for 17 h at 65 °C in a rotating Agilent hybridization oven. After hybridization, microarrays were washed 1 min at room temperature with the GE Wash Buffer 1 (Agilent) and 1 min with the 37 °C GE Wash buffer 2 (Agilent), then dried immediately by brief centrifugation. Slides were scanned immediately after washing on the Agilent SureScan Microarray Scanner G4900DA (Agilent Technologies) using one color scan setting for 8 × 60k array slides. (Scan Area 71 × 21.6 mm; Scan resolution 10 um; Dye channel is set to Green; and Green PMT is set to 100%). The scanned images were analyzed with Feature Extraction Software 9.1 (Agilent) using default parameters (protocol GE1_1200_Jun14 (Read Only) and Grid: 072363_D_F_20150612) to obtain background subtracted and spatially detrended Processed Signal intensities. Features flagged in Feature Extraction as Feature Non-uniform outliers were excluded. The raw data were analyzed by GeneSpring Ver. 14.9.1 (Agilent, CA, USA), and the signal value was Normalized. The biological processes of upregulated DEGs were analyzed by Gene Ontology (GO), and the results were presented as bubble plots generated using R, version 3.6.2. Adjusted *p* < 0.05 was considered as the cutoff value. The top 10 hub genes were analyzed by the protein-protein interaction network (PPI) using STRING (http://string-db.org/, accessed date: 19 May 2021) and Cytoscape 3.7.2 (cytohubba) (Cytoscape Consortium, Boston, MA, USA). A combined score > 0.4 was considered as the cutoff value, and the hub genes were selected according to the degree. Finally, the hub genes that were differentially expressed in G-SCs and P-SCs were identified by a heatmap.

### 2.13. Statistical Analysis

The statistical analysis in this study was performed using GraphPad Prism 9 (GraphPad Software Inc., San Diego, CA, USA). Data are shown as the mean ± standard deviation (s.d.). The comparison between the two groups was performed by Student’s *t*-test, and the comparison between two variables was performed by two-way ANOVA. Results with *p* < 0.05 were considered as statistically significant.

## 3. Results

### 3.1. Morphology and Proliferation of Normal Stromal Cells (G-SCs, P-SCs, and HDFs)

The morphology of G-SCs, P-SCs, and HDFs was examined by Giemsa staining, which indicated that G-SCs, P-SCs, and HDFs had a similar shape and were either spindle- or dendritic-shaped. The morphology of G-SCs was slightly smaller than that of the other two cell types. (Figure 1A). Furthermore, the cellular component of G-SCs, P-SCs, and HDFs were tested by IF, and G-SCs, P-SCs, and HDFs were vimentin-positive and AE1/3-negative, which suggested that these three types of cells did not contain epithelial cells (Figure 1B). Finally, the cell viability of G-SCs, P-SCs, and HDFs was tested by MTS assay at 1, 3, and 5 days, which confirmed that these three types of cells were viable. P-SCs showed the highest proliferation ability, followed by G-SCs (Figure 1C). These results indicated that G-SCs, P-SCs, and HDFs had spindle or dendritic morphology, similar to normal fibroblast cells and suitable cell viability and were devoid of epithelial cells.

### 3.2. G-SCs and P-SCs Influenced the Differentiation of OSCC Cells In Vitro

Both Giemsa staining and IF were performed to test the effect of G-SCs and P-SCs on the differentiation of OSCC cells in vitro. The HSC-2 only group showed strong adhesion between cancer cells and formed large cancer nests. The sizes of the cancer nests of both HSC-2+G-SCs and HSC-2+P-SCs were smaller than that of the HSC-2 only group and HSC-2+HDFs group. HSC-2+G-SCs had a higher cell density in cancer nests than in HSC-2+P-SCs. In HSC-2+ HDFs, large spinocellular-like cancer cells were observed (Figure 2A).

In the HSC-3 only group, cancer cell morphology was heterogeneous and showed strong cell atypia. HSC-3+G-SCs and HSC-3+P-SCs formed small cancer nests, whereas the HSC-3 only group did not form cancer nests. In HSC-3+G-SCs, both the cancer cells and stromal cells densely proliferated together, compared to HSC-3+P-SCs. HSC-3+ HDFs formed larger cancer nests than HSC-3+G-SCs and HSC-3+P-SCs (Figure 2B).

### 3.3. G-SCs and P-SCs Promoted the Proliferation of OSCC Cells In Vitro

The effect of G-SCs and P-SCs on the proliferation of OSCC cells was first tested by MTS assay. In the HSC-2 experiment, the OD value of HSC-2+G-SCs was slightly higher than that of HSC-2+P-SCs, but dramatically higher than that of HSC-2+HDFs. In addition, the OD values of these three groups of cells increased on day 3, but decreased significantly on day 5 (Figure 3A).

In the HSC-3 experiment, the OD value of HSC-3+G-SCs was higher than that of HSC-3+P-SCs, but significantly higher than that of the HSC-3+HDFs group. In addition, the OD values of these three groups of cells increased on day 3 and decreased dramatically on day 5 (Figure 3B). Therefore, G-SCs, P-SCs, and HDFs promoted the proliferation of OSCC cells in vitro.

Furthermore, Ki-67 immunohistochemical staining was performed to examine the effect of G-SCs and P-SCs on the proliferation of OSCC cells in vitro (Figure 3C,D). The percentage of Ki-67 positive cells in HSC-2+G-SCs, HSC-2+P-SCs, and HSC-2+HDFs groups was dramatically higher than that in the HSC-2 only group, and there was little difference between the HSC-2+G-SCs and HSC-2+P-SCs groups (Figure 3E). The percentage of Ki-67 positive cells in HSC-3+P-SCs was slightly higher than that in HSC-3+G-SCs and dramatically higher than that in the HSC-3+HDFs and HSC-3 only groups. In addition, there was little difference between the HSC-3+HDFs and HSC-3 only groups (Figure 3F). Normal stromal cells promoted the proliferation of any type of OSCC cells, and gingival stromal cells mostly affected the proliferative activity.

### 3.4. G-SCs and P-SCs Promoted the Invasion of OSCC Cells In Vitro

The effect of G-SCs and P-SCs on the invasion of OSCC cells in vitro was examined by the transwell invasion assay (Figure 4A,C). The cell invasion number in the HSC-2+G-SCs group was slightly higher than that in HSC-2+P-SCs, and was dramatically higher than that in the HSC-2+HDFs and HSC-2 only groups. In addition, the cell invasion number in the HSC-2+HDFs group was slightly higher than that in the HSC-2 (10,000 cells) group (Figure 4B). The cell invasion number in HSC-3+P-SCs was the highest, followed by the HSC-3+G-SCs group. There was little difference between the HSC-3 and HSC-3+HDFs groups (Figure 4D). These data suggest that G-SCs and P-SCs could promote the invasion of OSCC cells in vitro.

### 3.5. G-SCs and P-SCs Promoted the Migration of OSCC Cells In Vitro

The effect of G-SCs and P-SCs on the migration of OSCC cells in vitro was examined by the transwell migration assay (Figure 5A,C). The cell migration number in HSC-2+G-SCs was slightly higher than that in HSC-2+P-SCs, and dramatically higher than that in the HSC-2+HDFs and HSC-2 only groups. Furthermore, there was little difference between the HSC-2 only group and the HSC-2+HDFs group (Figure 5B). The cell migration number in HSC-3+G-SCs was the highest, followed by the HSC-3+P-SCs group. There was little difference between the HSC-3 only group and the HSC-3+HDFs group (Figure 5D). Therefore, both G-SCs and P-SCs could promote the migration of OSCC cells in vitro, and G-SCs promoted better migration than P-SCs.

### 3.6. G-SCs and P-SCs Influence the Histological Differentiation of OSCC In Vivo

The effect of G-SCs and P-SCs on the differentiation of OSCC in a xenograft mouse model was examined by HE staining. HSC-2+HDFs and HSC-2 only groups had keratinized regions; thus, their histological differentiation indicated that they were well-differentiated OSCC. In addition, the size of cancer nests in HSC-2+G-SCs and HSC-2+P-SCs was smaller than that in the HSC-2 only group and the HSC-2+HDFs group, and their cancer nests did not show keratinization. Thus, their histological differentiation indicated that they were moderately differentiated OSCC (Figure 6A).

The size of cancer nests in HSC-3+G-SCs and the HSC-3 only group was small, and histological differentiation of these two groups of cells indicated moderately to poorly differentiated OSCC (Figure 6B). In contrast, the histological differentiation of HSC-3+P-SCs and HSC-3+HDFs showed keratinized regions over the skull (Figure 6C,E). However, in the skull, HSC-3+P-SCs and HSC-3+HDFs had small cancer nests and showed moderately to poorly differentiated OSCC (Figure 6D,F). Therefore, in HSC-2 experiments, both G-SCs and P-SCs inhibited the differentiation of OSCC cells. However, in HSC-3 experiments, while G-SCs and P-SCs inhibited the differentiation of OSCC, P-SCs, in addition, could promote the differentiation of OSCC upon contact with bone tissue.

### 3.7. G-SCs and P-SCs Influenced the Proliferation of OSCC In Vivo

Ki-67 immunohistochemical staining was performed to examine the effect of G-SCs and P-SCs on the proliferation of OSCC cells in vivo (Figure 7A,B). The percentage of Ki-67 positive cells in the HSC-2+G-SCs group was slightly higher than that in the HSC-2+P-SCs group, and dramatically higher than that in the HSC-2 only and HSC-2+HDFs groups (Figure 7C). Thus, both G-SCs and P-SCs could promote cancer cell proliferation.

The percentage of Ki-67-positive cells in HSC-3+G-SCs was the highest among the other groups. Both HSC-3+P-SCs and HSC-3+HDFs had a higher Ki-67 positive rate in the invasion region than in the HSC-3 only group. However, the average Ki-67 positive rate, including the keratinized region, was slightly lower than that in the HSC-3 only group (Figure 7D).

### 3.8. G-SCs and P-SCs Promoted Bone Invasion by OSCC In Vivo

The effect of G-SCs and P-SCs on bone resorption in OSCC was first tested by HE staining. The degree of bone resorption in HSC-2+G-SCs and HSC-2+P-SCs was more severe than that in the HSC-2 only group and the HSC-2+HDFs group, in which no bone resorption was observed (Figure 8A). Furthermore, the active multi-nucleated osteoclasts on the bone resorption surface were stained for TRAP to determine the effect of G-SCs and P-SCs on bone invasion by HSC-2 cells in vivo (Figure 8B). The number of active multi-nucleated osteoclasts in the HSC-2+G-SCs group was dramatically higher than that in the HSC-2+P-SCs and HSC-2+HDFs groups, and the HSC-2 only group. There was little difference between the HSC-2 group and HSC-2+HDFs group (Figure 8E). In addition, the degree of bone resorption in the HSC-3+G-SCs and HSC-3+P-SCs groups was more severe than that in the HSC-3 only and HSC-3+HDFs groups (Figure 8C). Meanwhile, the number of active multi-nucleated osteoclasts in HSC-3+G-SCs was the highest, followed by the HSC-3+P-SCs groups. There was little difference between the HSC-3 and HSC-3+HDFs groups (Figure 8D,F). These results demonstrated that both G-SCs and P-SCs promoted bone invasion by OSCC in vivo and that G-SCs could promote better bone invasion than P-SCs.

### 3.9. CDK1, BUB1B, TOP2A, DLGAP5, BUB1, and CCNB2 Have Great Potential to Cause the Different Effects of G-SCs and P-SCs, Hence Resulting in the Progression of OSCC

The upregulated genes in P-SCs compared to G-SCs were selected by microarray analysis. The biological processes of upregulated DEGs in P-SCs were analyzed by GO enrichment analysis, which indicated that these upregulated DEGs were mainly enriched in cancer-associated biological processes, such as the cell cycle, cell proliferation, cell differentiation, and cell migration (Figure 9A). Furthermore, the upregulated DEGs in P-SCs associated with cancer biological processes were selected, and PPI network analysis was performed to identify the hub genes. The analysis demonstrated that *CDK1*, *BUB1B*, *TOP2A*, *DLGAP5*, *BUB1*, *PLK1*, *AURKB*, *CCNB1*, *CCNB2*, and *AURKA* were the hub genes (Figure 9B). Finally, the differentially expressed hub genes were analyzed by heatmap, which demonstrated that *CDK1*, *BUB1B*, *TOP2A*, *DLGAP5*, *BUB1*, and *CCNB2* were differentially expressed in G-SCs and P-SCs (Figure 9C). Collectively, these data demonstrated that *CDK1, BUB1B, TOP2A, DLGAP5, BUB1,* and *CCNB2* have a greater potential to cause the different effects of G-SCs and P-SCs, thereby resulting in the progression of OSCC.

## 4. Discussion

Our data are the first to demonstrate that the normal stroma (G-SCs and P-SCs) is directly associated with changes in the biological characteristics of the OSCC tumor parenchyma, such as proliferation, invasion, and morphology, both in vitro and in vivo. Generally, OSCC develops due to the accumulation of abnormal genes, and the biological characteristics of OSCC are controlled by the tumor parenchyma. Honglin et al. have reported that the stromal stiffness regulates the malignancy degree in pancreatic cancer, thus that the tumor stroma affects the character of the tumor parenchyma [14]. However, there are no reports that the tumor stroma controls the biological character of the tumor parenchyma. Our research results indicate that the tumor parenchyma may be controlled by the stroma directly, not by the accumulation of the genetic abnormality.

### 4.1. Effect of G-SCs and P-SCs for OSCC Comparing to HDF

In contrast to the above findings, HDFs had little effect on the biological characteristics of OSCC. Cutaneous squamous cell carcinoma (CSCC) has a better prognosis than OSCC, and CSCC rarely invades deep into the skin tissue [15,16,17], whereas OSCC has significant invasion ability [18]. Although both CSCC and OSCC develop from the epithelium in a similar manner, their infiltration ability and prognosis are quite different. We suggest three reasons for this result. The first reason is that the surrounding environment, which is in contact with the cancer cells, is different. G-SCs and P-SCs are neural crest-derived cells (ectoderm), whereas HDFs are mesoderm-derived cells. Second, the connective tissue derived from the oral cavity has a higher affinity and coordination with squamous cell carcinoma than fibroblasts derived from the skin connective tissue. Stephen Paget proposed the “seed and soil” theory of cancer. He proposed that tumor cells (the seeds) have a specific affinity for specific organs (the soil), and metastasis would occur only if the seed and soil were compatible [19,20,21]. In fact, OSCC frequently metastasizes to the submandibular lymph nodes, cervical lymph nodes, and lungs; however, OSCC metastasis to other organs is relatively rare [22,23]. Finally, HDFs are composed of a single type of cells and do not contain stem cells, unlike G-SCs and P-SCs which are heterogeneous cell populations and contain stem cells. The interaction between tumor-secreted factors and mesenchymal stem cells contributes to tumor progression via several mechanisms, such as differentiation into other pro-tumorigenic components of the TME, suppression of immune response, promotion of angiogenesis, and enhancement of the epithelial to mesenchymal transition (EMT) [24,25,26,27,28,29].

### 4.2. Effect of the Stromal Cells on the Proliferative, Invasive, and Migration Abilities of OSCC

Next, we compared the experimental results between the G-SC and P-SC groups. The proliferation of both HSC-2+G-SCs and HSC-2+P-SCs was significantly higher than the HSC-2 alone group in vitro and in vivo, and G-SCs were more effective than P-SCs in enhancing the proliferative activity of HSC-2. However, in vitro and in vivo tumor cell proliferation experiments using HSC-3 showed that both G-SCs and P-SCs had no significant effect on the proliferative activity of HSC-3 cells in vitro. In vivo, the proliferative activity of cancer cells in HSC-3+P-SCs was higher than that in HSC-3+G-SCs as observed in the invasion area. However, the proliferative activity of cancer cells in HSC-3+P-SCs was significantly suppressed in the keratinized area. Therefore, based on the average rate of proliferation in HSC-3+P-SCs, the effect of HSC-3 on proliferative activity was offset, and there was no effect on cancer proliferation. These data suggest that both the G-SC and P-SC groups promoted the proliferation of well-differentiated cancer cells and that G-SCs affected the proliferation of poorly differentiated cancer cells compared to the P-SC group. Recent studies have demonstrated that transforming growth factor-β (TGF-β), which activates the infiltration ability of cancer cells, is produced by many cancer cells [30,31,32]. It has also been reported that fibroblast growth factor 2 (FGF-2) suppresses the action of TGF-β on cancer-associated fibroblasts (CAFs) to promote tumor formation [33]. Akatsu et al. reported that TGF-β and FGF-2 antagonized or cooperated with each other to control the formation of cancer-restraining CAFs and cancer-promoting CAFs, respectively, and that TGF-β and FGF-2 determine CAF characteristics in the tumor microenvironment [34]. Endogenous FGF-2 is widely expressed in periodontal tissues, such as the periodontal ligament, gingival connective tissue, and gingival epithelial cell gap. FGF-2 has been reported to be involved in bone tissue formation [35,36], and its expression is higher in the periodontal ligament, which has a bone-forming ability, than in the gingival connective tissue. TGF-β is expressed more in poorly differentiated OSCC than in well-differentiated OSCC. Therefore, our data suggest that HSC-3 suppresses the proliferative activity rather than activates it since HSC-3 secretes more TGF-β than HSC-2.

### 4.3. Effect of the Stromal Cells on the Morphology and Differentiation Degree of OSCC

In this study, HSC-2, which is a well-differentiated OSCC cell line, was changed to moderately differentiated OSCC in the HSC-2+G-SCs and HSC-2+P-SCs groups. However, although the histological differentiation of the HSC-2+G-SCs and HSC-2+P-SCs groups were the same moderately differentiated type, the HSC-2+P-SCs group formed larger tumor nests than HSC-2+G-SCs, and showed tumor morphology with increased cell adhesion. In co-culture experiments using HSC-3, which is moderately differentiated OSCC, histological differentiation of the HSC-3+G-SCs group showed moderate to poorly differentiated OSCC. On the other hand, in the HSC-3+P-SCs group, large cancer nest formation with keratinization was observed in most parts, and histological differentiation revealed moderate to poorly differentiated OSCC from the bone invasion part to the deep part.

Hsieh et al. reported that elastin, which is the main component of elastic fibers present in the connective tissue, is involved in the keratinization of the epithelium in the oral mucosa, and that the addition of elastin during organ culture of keratinized gingiva altered keratinized gingiva to non-keratinized gingiva [37]. Only oxytalan fibers that were delicate in nature were present in the periodontal ligament, and there were no elastin-deposited elastic fibers. On the other hand, in the gingival lamina propria, elastin deposition was observed. Moreover, not only oxytalan fibers but also elastic fibers, such as elaunin fibers, were detected [38].

In addition, Green et al. and Itoh et al. reported that ES cells and induced pluripotent stem (iPS) cells could be stimulated to form keratinocytes by inducing differentiation using bone morphogenetic protein 4 (BMP-4) and retinoic acid [39,40]. It has been reported that the localization of osteopontin, osteocalcin, bone morphogenetic protein 2 (BMP-2), and BMP-4 in the periodontal tissue is stronger in the periodontal ligament than in the gingival connective tissue [41]. Thus, it is considered that the periodontal ligament promotes better keratinization compared to the gingival connective tissue.

### 4.4. Microarray Analysis of Stromal Cells

*Cyclin dependent kinase 1 (CDK1)* plays a significant role in the cell cycle process in cancers [42]. A recent study also indicated that an enhancer of zeste 1 polycomb repressive complex 2 subunit (EZH1) could be regulated by *CDK1*, which promotes the osteogenic differentiation of human mesenchymal stem cells [43,44]. *CDK1* expression is also associated with the proliferation of early osteoblastic cells [45]. However, the role of *CDK1* in the crosstalk between G-SCs/P-SCs and OSCC has been poorly investigated. A recent study demonstrated that *BUB1 mitotic checkpoint serine/threonine kinase B (BUB1B)* could promote the proliferation of prostate cancer [46]. *BUB1B* also promotes the progression of hepatocellular carcinoma by activating the mammalian target of the rapamycin complex 1 (mTORC1) signaling pathway [47]. However, the role and function of *BUB1B* in the crosstalk between G-SCs/P-SCs and OSCC have been poorly studied. *DNA topoisomerase II alpha (TOP2A)* is a nuclear protein associated with cell division [48]. Thus, *TOP2A* is closely associated with cancer proliferation [49]. However, little is known about the role and function of *TOP2A*. *DLG-associated protein 5 (DLGAP5)* is closely associated with the prognosis of many types of cancers [50,51]. *Nucleolar and Spindle Associated Protein1 (NUSAP1)* inhibits the proliferation of invasive breast cancer cells by regulating DLGAP5 expression [52]. However, *DLGAP5* had little effect on the crosstalk between stromal cells and OSCC cells. *BUB1 mitotic checkpoint serine/threonine kinase (BUB1)* is overexpressed in pancreatic ductal adenocarcinoma, which could be a useful parameter to predict the poor prognosis of pancreatic ductal adenocarcinoma (PDAC) [53]. Overexpression of *BUB1* influences the progression of multiple myeloma by promoting mitotic segregation errors and chromosomal instability [54]. Therefore, *BUB1* plays a significant role in the progression of many types of cancers in different ways, but it has been poorly investigated in the crosstalk between stromal cells and cancer cells. *Cyclin B2 (CCNB2)* is a member of the cyclin family that is overexpressed in many types of cancers. A recent study suggested that miR-335-5p could regulate the cell cycle and metastasis of lung cancer by targeting *CCNB2* [55]. miR-205 inhibits the proliferation and migration of thyroid cancer cells by targeting *CCNB2* [56]. Therefore, *CCNB2* is a potential therapeutic target for thyroid cancer. However, its role in the crosstalk between stromal cells and OSCC is poorly investigated.

Taken together, this study showed that gingival connective tissue and the periodontal ligament may directly affect the biological character of the tumor parenchyma. However, this study has a potential limitation. In the future, similar experiments using the connective tissue of the tongue will need to be conducted. Although there is little environmental change due to tumor cell infiltration, a similar experimental system should be performed using the connective tissue of the tongue and compared to these experiment results because the OSCC derived from the tongue is the most common in oral cancer.

## 5. Conclusions

In conclusion, both the G-SC and P-SC groups promoted the proliferation of well-differentiated cancer cells and that G-SCs affected the proliferation of poorly differentiated cancer cells compared to the P-SC group. In addition, P-SCs promoted the differentiation degree of moderately differentiated OSCC compared to G-SCs. *CDK1*, *BUB1B*, *TOP2A*, *DLGAP5*, *BUB1*, and *CCNB2* have great potential to cause the different effects of G-SCs and P-SCs, hence resulting in the progression of OSCC. Based on the results derived from multiple approaches, we propose that the biological characteristics of OSCC cells are altered upon contact with the surrounding tissues, such as gingival connective tissue and the periodontal ligament. Finally, the potential molecular mechanism involved in normal stroma-induced change in the biological characteristics of OSCC cells warrants further investigation.

## Figures and Tables

**Figure 1 cancers-13-03491-f001:**
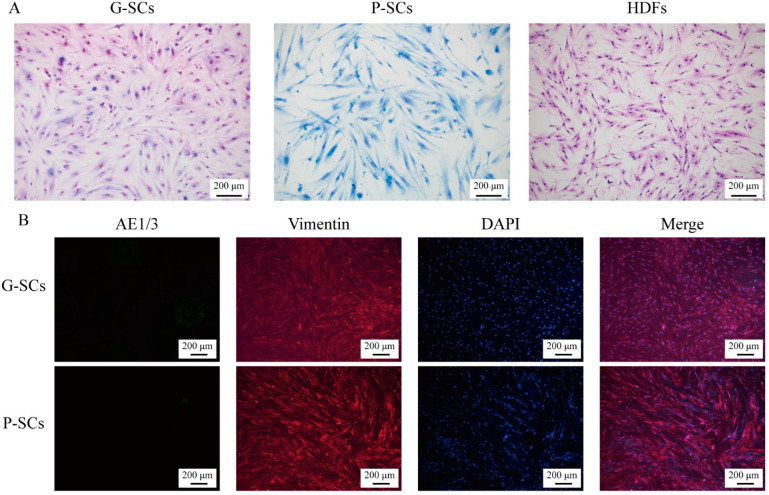
The cell morphology, cell viability, and cellular component of G-SCs, P-SCs, and HDFs were confirmed. (**A**) The cell morphology of G-SCs, P-SCs, and HDFs was confirmed by Giemsa staining. (**B**) The cellular component of G-SCs, P-SCs, and HDFs was confirmed by IF. (**C**) The cell viability of G-SCs, P-SCs, and HDFs was examined by MTS assay. Data are shown as the mean ± s.d. (s.d., standard deviation), *n* = 4. Statistical analysis was performed by two-way ANOVA, **** indicates *p* < 0.0001.

**Figure 2 cancers-13-03491-f002:**
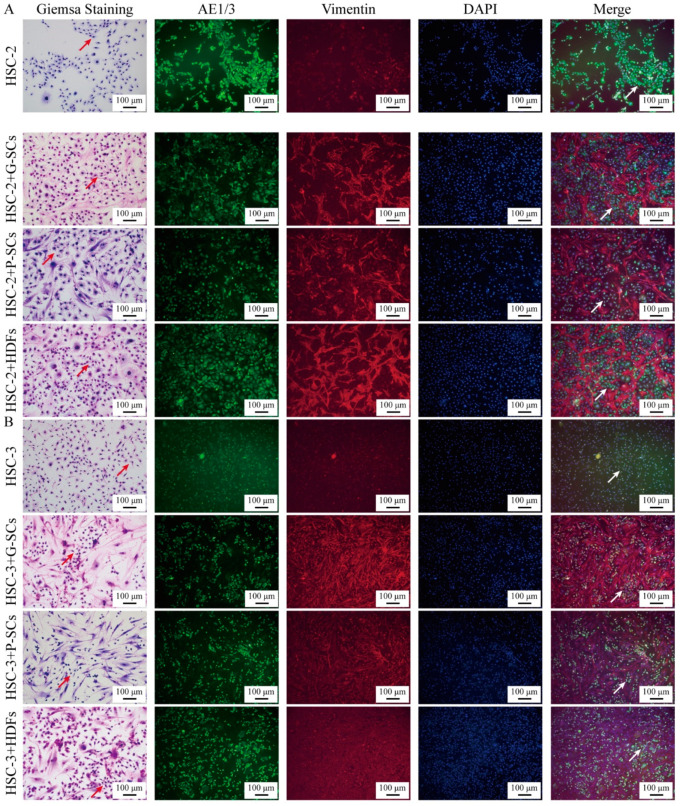
G-SCs and P-SCs inhibited the differentiation of OSCC cells in vitro. (**A**) The effect of G-SCs and P-SCs on the differentiation of HSC-2 was examined by both Giemsa staining and IF. (**B**) The effect of G-SCs and P-SCs on the differentiation of HSC-3 was examined by both Giemsa staining and IF.

**Figure 3 cancers-13-03491-f003:**
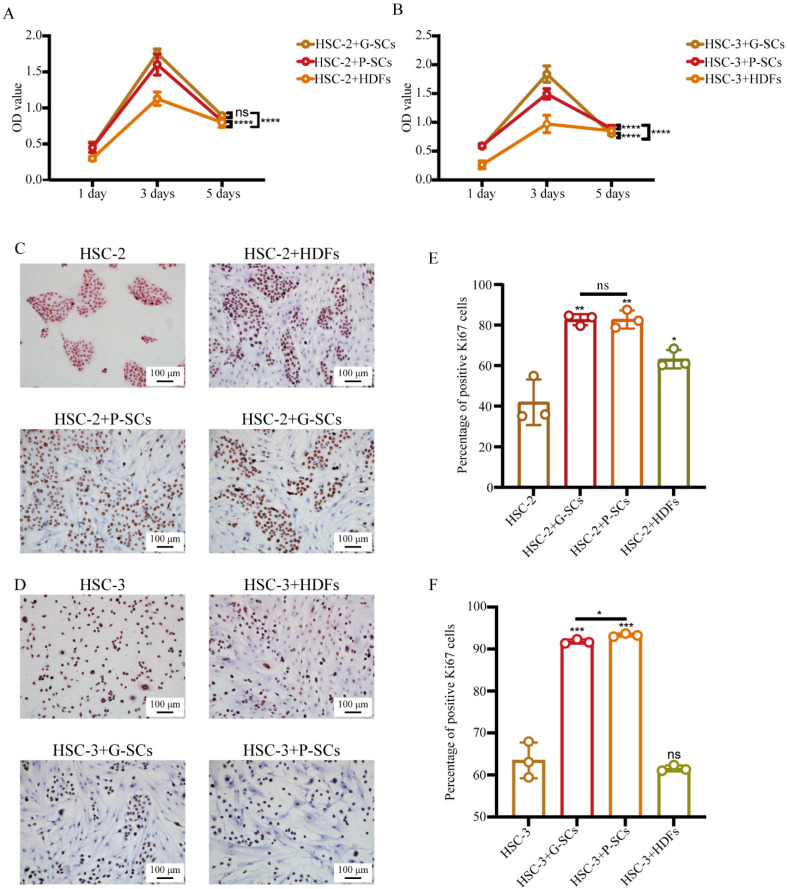
G-SCs and P-SCs promoted the proliferation of OSCC cells in vitro. (**A**,**B**) The effect of G-SCs/P-SCs/HDFs on the proliferation of HSC-2/3 in vitro was tested by MTS assay. Data are shown as the mean ± s.d. (s.d., standard deviation), *n* = 4. Statistical analysis was performed by two-way ANOVA, ns indicates *p* > 0.05, **** indicates *p* < 0.0001. (**C**,**D**) Ki-67 IHC staining was performed to examine the effect of G-SCs/P-SCs/HDFs on the proliferation of HSC-2/3 in vitro. (**E**,**F**) Quantification of Ki-67 positive cells in different groups. Data are shown as the mean ± s.d. (s.d., standard deviation), *n* = 3. Each group compared with HSC-3 group and statistical analysis was performed by Student’s *t*-test, ns indicates *p* > 0.05, * indicates *p* < 0.05, ** indicates *p* < 0.01, *** indicates *p* < 0.001.

**Figure 4 cancers-13-03491-f004:**
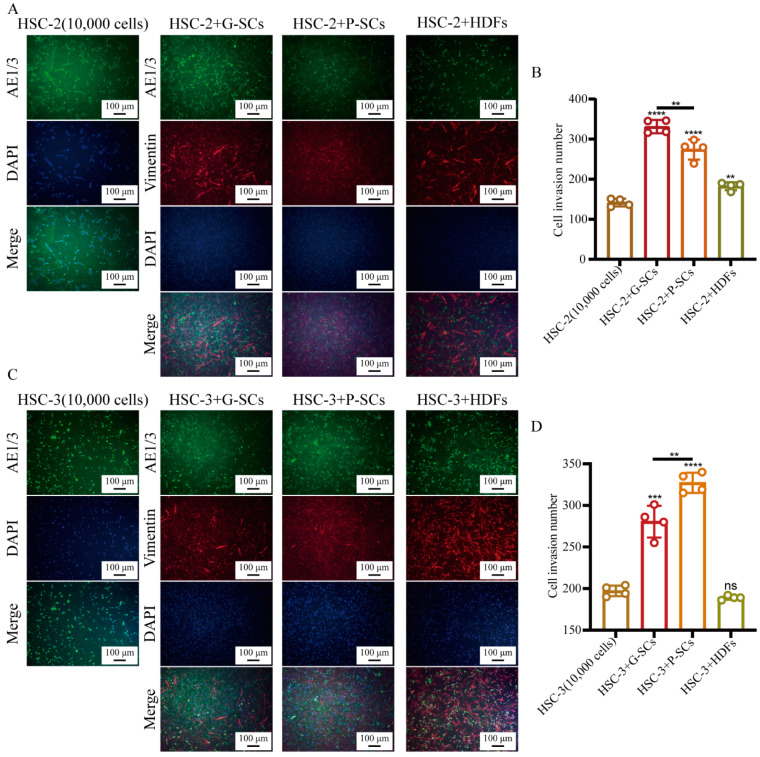
G-SCs and P-SCs promoted the invasion of OSCC cells in vitro. (**A**,**C**) The effect of G-SCs and P-SCs on the invasion of HSC-2 (**A**) and HSC-3 (**C**) was examined by transwell invasion assay. (**B**,**D**) Quantification of cell invasion number in different groups of HSC-2 (**B**) and HSC-3 (**D**). Data are shown as the mean ± s.d. (s.d., standard deviation), *n* = 4. Each group compared with HSC-2 (10,000 cells) and HSC-3 (10,000 cells) group, and statistical analysis was performed by Student’s *t*-test, ns indicates *p* > 0.05, ** indicates *p* < 0.01, *** indicates *p* < 0.001, **** indicates *p* < 0.0001.

**Figure 5 cancers-13-03491-f005:**
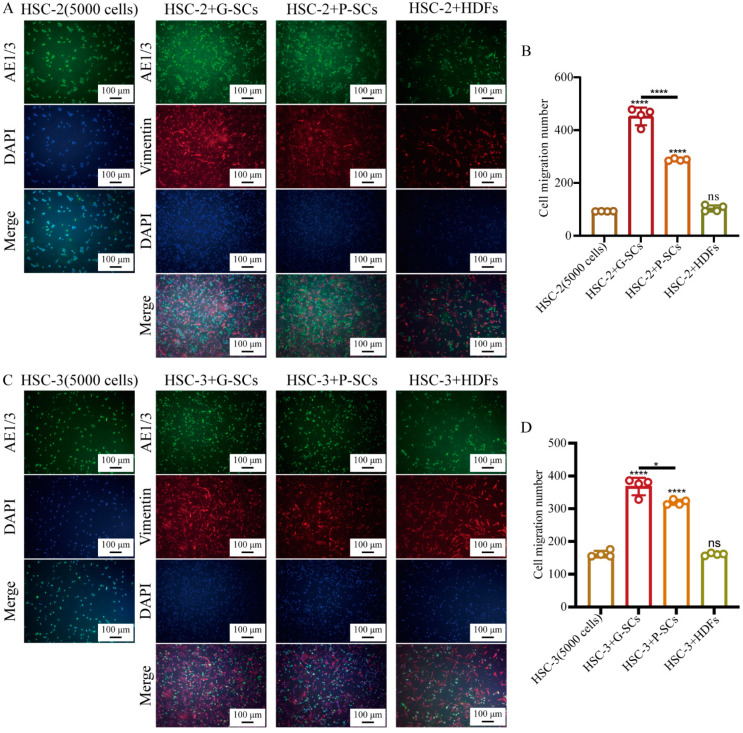
G-SCs and P-SCs promoted the migration of OSCC cells in vitro. (**A**,**C**) The effect of G-SCs and P-SCs on the migration of HSC-2 (**A**) and HSC-3 (**C**) was examined by transwell migration assay. (**B**,**D**) Quantification of cell migration in different groups of HSC-2 (**B**) and HSC-3 cells (**D**). Data are shown as the mean ± s.d. (s.d., standard deviation), *n* = 4. Each group compared with HSC-2 (5000 cells) and HSC-3 (5000 cells) group, and statistical analysis was performed by Student’s *t*-test, ns indicates *p* > 0.05, * indicates *p* < 0.05, **** indicates *p* < 0.0001.

**Figure 6 cancers-13-03491-f006:**
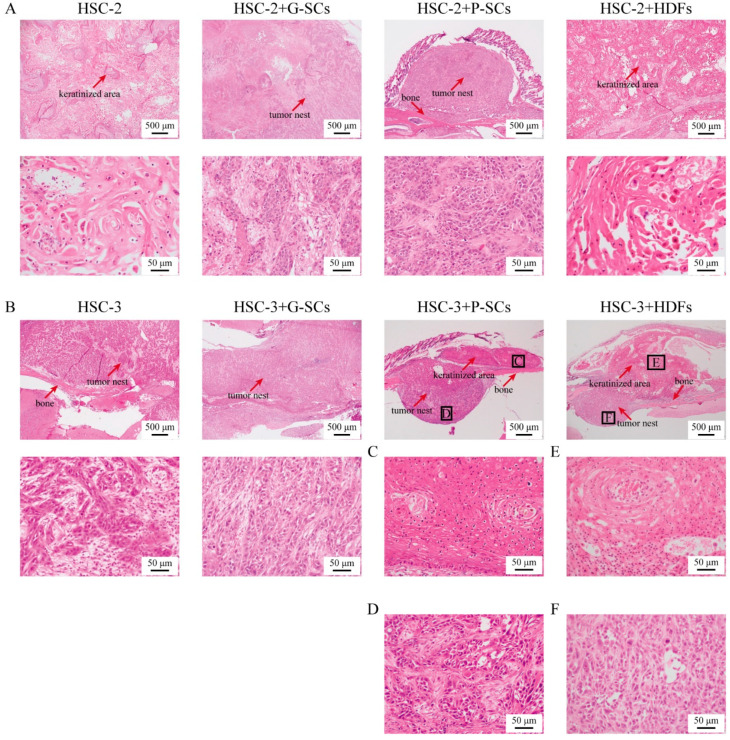
G-SCs and P-SCs inhibited the differentiation of OSCC in vivo. (**A**) The effect of G-SCs and P-SCs on the differentiation of HSC-2 in vivo was examined by HE staining. (**B**) The effect of G-SCs and P-SCs on the differentiation of HSC-3 in vivo was examined by HE staining. (**C**) The keratinized area of HSC-3+P-SCs. (**D**) The invasion area of HSC-3+P-SCs. (**E**) The keratinized area of HSC-3+HDFs. (**F**) The invasion area of HSC-3+HDFs.

**Figure 7 cancers-13-03491-f007:**
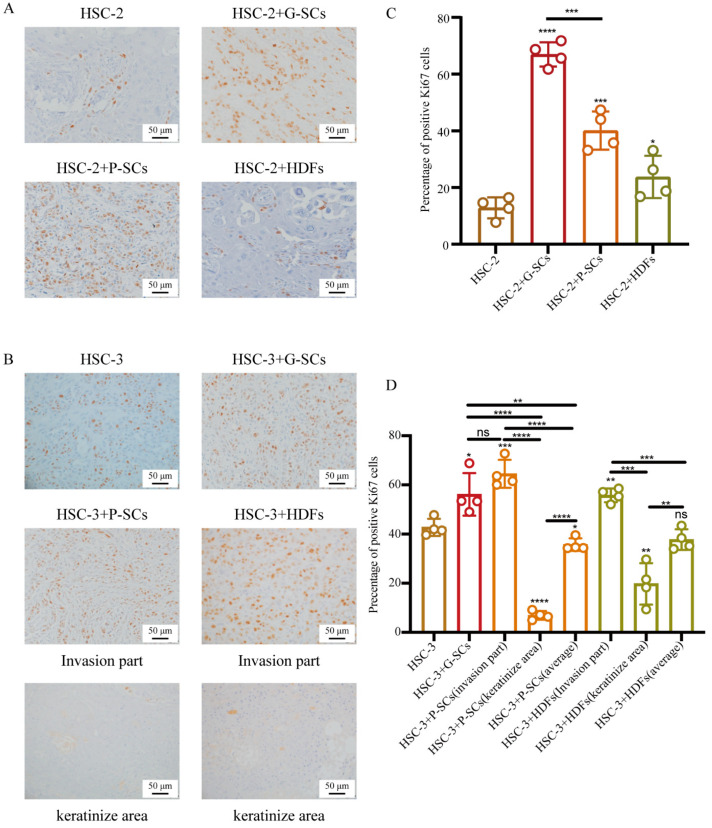
G-SCs and P-SCs influenced the proliferation of OSCC in vivo. (**A**,**B**) Ki-67 IHC staining was performed to examine the effect of G-SCs and P-SCs on the proliferation of HSC-2 (**A**) and HSC-3 (**B**) in vivo. (**C**,**D**) Quantification of Ki-67 positive cells in different groups of HSC-2 (**C**) and HSC-3 (**D**). Data are shown as the mean ± s.d. (s.d., standard deviation), *n* = 4. Each group compared with HSC-2 and HSC-3, respectively, and statistical analysis was performed by Student’s *t*-test, ns indicates *p* > 0.05, * indicates *p* < 0.05, ** indicates *p* < 0.01, *** indicates *p* < 0.001, **** indicates *p* < 0.0001.

**Figure 8 cancers-13-03491-f008:**
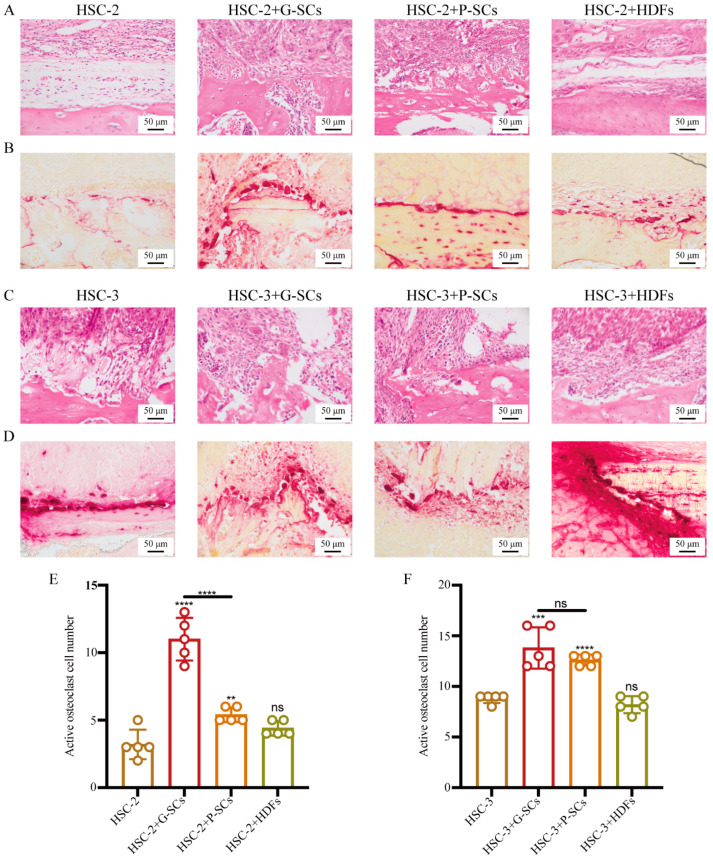
G-SCs and P-SCs promoted bone invasion by OSCC in vivo. (**A**,**C**) The effect of G-SCs and P-SCs on bone resorption of HSC-2 (**A**) and HSC-3 (**C**) in vivo was examined by HE staining. (**B**,**D**) The number of active multi-nucleated osteoclasts upon bone resorption was tested by TRAP staining to examine the effect of G-SCs and P-SCs on bone resorption of HSC-2 (**B**) and HSC-3 (**D**) in vivo. (**E**,**F**) Quantification of activated multi-nucleated osteoclast cell number in different groups of HSC-2 (E) and HSC-3 (F). Data are shown as the mean ± s.d. (s.d., standard deviation), *n* = 5. Each group compared with HSC-2 and HSC-3 group, respectively, and statistical analysis was performed by Student’s *t*-test, ns indicates *p* > 0.05, ** indicates *p* < 0.01, *** indicates *p* < 0.001, **** indicates *p* < 0.0001.

**Figure 9 cancers-13-03491-f009:**
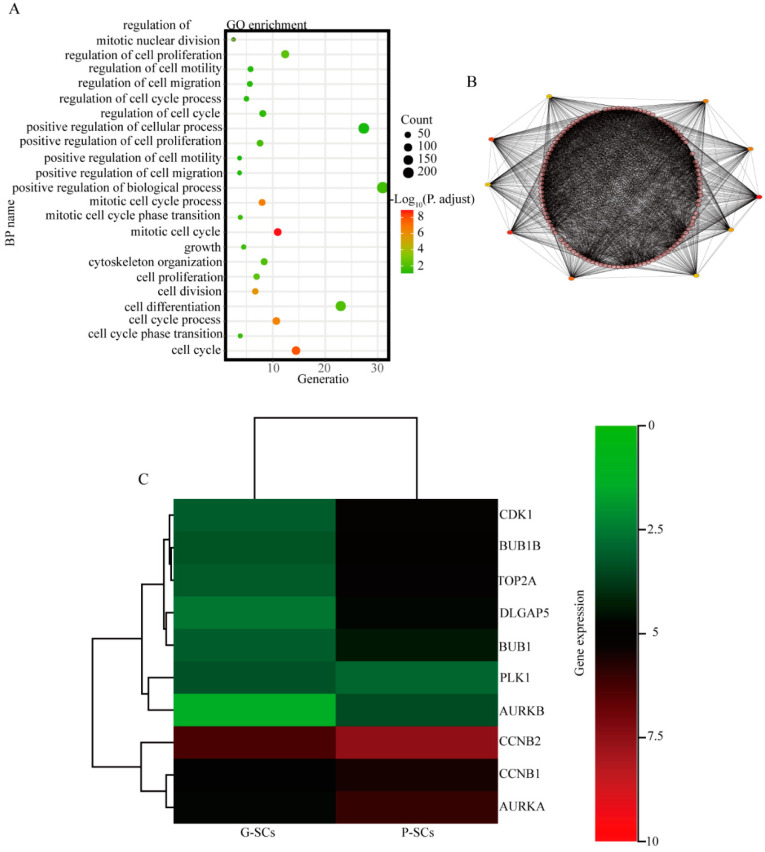
Identification of potential genes that cause different effects of G-SCs and P-SCs resulting in the progression of OSCC. (**A**) The biological processes of upregulated DEGs in P-SCs were analyzed by GO enrichment analysis (cancer-associated biological processes). (**B**) The hub genes in upregulated DEGs associated with cancer-related biological processes were analyzed by PPI network. (**C**) The hub genes differentially expressed in G-SCs and P-SCs were analyzed by heatmap.

## Data Availability

The data that support findings of this study are available from the corresponding author, e-mail, upon the reasonable request.

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
