# Peer review of "The Origin of Stroma Influences the Biological Characteristics of Oral Squamous Cell Carcinoma"

_cancers, 2021, doi:10.3390/cancers13143491_

Round 1
Reviewer 1 Report
In this paper, authors study the effects of normal gingival and periodontal ligament stromal cells on oral cancer. They find reduced differentiation and increased proliferation, migration, and invasion in oral squamous cell carcinoma (OSCC) cells cultured with gingival or periodontal ligament cells compared with those cultured with dermal fibroblasts and without stromal cells. In some assays, gingival cells appear to promote higher tumorigenesis and bone invasion than periodontal cells, but this evidence is equivocal when considering the paper as a whole. Finally, differentially expressed genes between the two cell types are identified using microarray analysis. Detailed feedback is below.
Introduction: clear and concise overall.
Methods
- In the introduction, authors compare oral cancer in tongue vs. gingiva and periodontal tissue, which differ from tongue in tumor spread and types of tissue that are invaded. However, in experiments, gingiva and periodontal stromal cells were compared to each other and to dermal fibroblasts. Tongue lamina propria fibroblasts may have been a more apt comparison to support the claim in the paper’s title. This should be discussed as a limitation or the title should be changed.
- Section 2.1: Sufficient detail is provided about periodontal ligament dissection but minimal detail is provided about gingival cells. How were these isolated? How many patients were these collected from? How were gingival and periodontal patient samples determined to be disease-free?
- Section 2.5: How long were cells co-cultured before Ki67 staining? Was the counterstain hematoxylin?
- Section 2.8: For xenograft studies, please describe location on the head where cells were injected and depth of injection.
- Section 2.12: Please describe the methods for RNA isolation and microarray analysis, the platform used, etc.
Results
- Section 3.2/Figure 2:
- Arrows or other indicators to identify spinocellular-like cells would be helpful.
- It is not clear what is meant by “the cancer cells and stromal cells acted in a coordinated manner” (lines 268-269).
- Section 3.3/Figure 3: Describing effects as “significantly” different among groups (line 280, line 283) implies a statistical test with p<.05, but no test results are provided.
- Figure 3-5 and 7 captions: Please describe which groups were compared in pairwise tests.
- Section 3.4/Figure 4:
- Images shown for HSC-2+G-SCs and HSC-2+P-SCs are identical, likely a copy-paste error.
- Group differences described as “little” are statistically significant and appear as large as some other differences discussed as significant, considering that the y-axes between the two graphs differ.
- Figure 6:
- This caption is incomplete. 6C-F are not described.
- Labeling bone and cancer nests would be helpful.
- Figure 7D: It would be helpful if colors were consistent with the rest of the paper.
- Section 3.8/Figure 8:
- It is not clear why authors argue that gingival stromal cells promoted better invasion than periodontal cells, since bone disappeared in the periodontal cell condition.
- The methods state that TRAP stained cells were counted, but those quantitative data are not presented.
- Lines 392-394 are almost identical to a sentence earlier in the paragraph.
- Section 3.9/Figure 9: I am not convinced by the microarray analysis. The cutoff value seems liberal and it seems that only a single patient’s cells were studied for each cell type. Also, if these were isolated from different patients, individual differences in gene expression could mask differences by cell origin.
Discussion
- Authors do not quite compare their results to the literature. Are there other areas of the body where origin of stromal cells affect tumor characteristics?
- The discussion proposes many potential explanations for the results. However, it is not written sufficiently clearly. Saying that results suggest epigenetic changes and differences in growth factors and elastic fibers, for example, is too strong when all of these factors are testable but were not tested in this paper.
- The conclusion is appropriately conservative based on what was done.
Stylistic
- Please consistently spell out acronyms and initialisms on first use. (ECM, IF, MTS, HE, IHC, TRAP, EMT, FGF-2, ES and iPS cells, EZH1, likely others that I missed.)
- There are many typographical errors in the discussion. The entire paper would benefit from proofreading.
Author Response
Reviewer1
In this paper, authors study the effects of normal gingival and periodontal ligament stromal cells on oral cancer. They find reduced differentiation and increased proliferation, migration, and invasion in oral squamous cell carcinoma (OSCC) cells cultured with gingival or periodontal ligament cells compared with those cultured with dermal fibroblasts and without stromal cells. In some assays, gingival cells appear to promote higher tumorigenesis and bone invasion than periodontal cells, but this evidence is equivocal when considering the paper as a whole. Finally, differentially expressed genes between the two cell types are identified using microarray analysis. Detailed feedback is below.
Introduction: clear and concise overall.
Response: The introduction part has been modified clearly and concisely. Please check it.
Methods
In the introduction, authors compare oral cancer in tongue vs. gingiva and periodontal tissue, which differ from tongue in tumor spread and types of tissue that are invaded. However, in experiments, gingiva and periodontal stromal cells were compared to each other and to dermal fibroblasts. Tongue lamina propria fibroblasts may have been a more apt comparison to support the claim in the paper’s title. This should be discussed as a limitation or the title should be changed.
Response: This limitation has been discussed in the Discussion part (Line 820-827). Please check it.
Section 2.1: Sufficient detail is provided about periodontal ligament dissection but minimal detail is provided about gingival cells. How were these isolated? How many patients were these collected from? How were gingival and periodontal patient samples determined to be disease-free?
Response: We have added to the text in Materials and Methods part-section 2.1. (Line96-100)
Section 2.5: How long were cells co-cultured before Ki67 staining? Was the counterstain hematoxylin?
Response: We conducted the co-culture for two days which added into the section 2.5 (Line 185), and the counterstain was Mayer’s hematoxylin (Line 197-198).
Section 2.8: For xenograft studies, please describe location on the head where cells were injected and depth of injection.
Response: The mixed cells were injected in mouse top of the head central area and the injection deep arrived at the lamina propria. All of these has been added into the materials and methods part-section 2.8 (Line 237), please check it.
Section 2.12: Please describe the methods for RNA isolation and microarray analysis, the platform used, etc.
Response: The relevant information for the microarray has been added into the materials and methods part-section 2.12 (Line 275-296), please check it.
Results
Section 3.2/Figure 2:
Arrows or other indicators to identify spinocellular-like cells would be helpful.
Response: the arrows have been added on the figure. 2. Please check it.
It is not clear what is meant by “the cancer cells and stromal cells acted in a coordinated manner” (lines 268-269).
Response: We have modified the text (Line 354-355).
Section 3.3/Figure 3: Describing effects as “significantly” different among groups (line 280, line 283) implies a statistical test with p<.05, but no test results are provided.
Response: The statistics analysis has been added on the Fig.3, please check it.
Figure 3-5 and 7 captions: Please describe which groups were compared in pairwise tests.
Response: The detail information for pairwise tests has been added into Figure3-5, 7 captions, please check it.
Section 3.4/Figure 4:
Images shown for HSC-2+G-SCs and HSC-2+P-SCs are identical, likely a copy-paste error.
Response: The picture for HSC-2+G-SCs and HSC-2+P-SCs in fig. 4 have been changed, please check it.
Group differences described as “little” are statistically significant and appear as large as some other differences discussed as significant, considering that the y-axes between the two graphs differ.
Response: The wrong description for this in Fig. 4 has been changed, please check it.
Figure 6:
This caption is incomplete. 6C-F are not described.
Response: The 6C-F has been added into the caption for Fig. 6. Please check it.
Labeling bone and cancer nests would be helpful.
Response: The bone and tumor nest were highlighted by the arrows, please check it.
Figure 7D: It would be helpful if colors were consistent with the rest of the paper.
The color of Fig. 7D has been changed to consistent with the rest of the paper, please check it.
Section 3.8/Figure 8:
It is not clear why authors argue that gingival stromal cells promoted better invasion than periodontal cells, since bone disappeared in the periodontal cell condition.
Response: In Fig.8C, the existing bone disappeared in HSC-3+G-SCs comparing to HSC-3+P-SCs group. Thus, we argue that gingival stromal cells promoted better invasion than periodontal cells.
In addition, we changed a suitable picture of HSC-3+P-SCs in Fig.8, please check it.
The methods state that TRAP stained cells were counted, but those quantitative data are not presented.
Response: The quantification of positive multi-nucleated osteoclast cell number has been added into Fig. 8, please check it.
Lines 392-394 are almost identical to a sentence earlier in the paragraph.
Response: the sentence from line 392 to 394 has been removed, please check it.
Section 3.9/Figure 9: I am not convinced by the microarray analysis. The cutoff value seems liberal and it seems that only a single patient’s cells were studied for each cell type. Also, if these were isolated from different patients, individual differences in gene expression could mask differences by cell origin.
Response: Thanks for your advice. It is quite difficult to get the normal gingival and periodontal ligament derived stromal cells from the heathy person or from the heath gingival and periodontal area from the patients. Therefore, we get these cells from one of heathy medical staff in our department. The limitation condition may result in the microarray data a little not standard and I think it can be accepted mainly because the microarray data is just the prediction instead of the definitive conclusion. As for the real role and function of these potential genes from microarray in the crosstalk between G-SCs/P-SCs and OSCC need to be further investigated by biological experiment.
Discussion
Authors do not quite compare their results to the literature. Are there other areas of the body where origin of stromal cells affect tumor characteristics?
The discussion proposes many potential explanations for the results. However, it is not written sufficiently clearly. Saying that results suggest epigenetic changes and differences in growth factors and elastic fibers, for example, is too strong when all of these factors are testable but were not tested in this paper.
The conclusion is appropriately conservative based on what was done.
Response: The discussion and conclusion has been modified according to the reviewer advice, please check it.
Stylistic
Please consistently spell out acronyms and initialisms on first use. (ECM, IF, MTS, HE, IHC, TRAP, EMT, FGF-2, ES and iPS cells, EZH1, likely others that I missed.)
There are many typographical errors in the discussion. The entire paper would benefit from proofreading.
Response: The full name for the abbreviation has been added in the article, please check it.

Reviewer 2 Report
The manuscript presented by Omori et al, entitled “ The Origin of Stroma Influences the Biological Characteristics of Oral Squamous Cell Carcinoma” describes the role of stromal cells derived from the gingival and periodontal ligament tissues in the proliferation, invasion, and migration of OSCC both in vitro and in vivo.
The authors present quite intricate data, that demonstrate for the first time that the normal stroma (G-SCs and P-SCs) is directly associated with changes in the biological characteristics of OSCC tumor parenchyma, such as proliferation, invasion, and morphology, both in vitro and in vivo. All this has the potential to be applied in the clinical practice as therapeutic strategy for OSCC.
Overall, their results indicated that both G-SCs and P-SCs could promote the progression of OSCC, which could be a potential regulatory mechanism in the progression of this malignancy.
In general, the material is well organized and easy to be followed and understood.
The results are clear and satisfactory.
The figure and the graphics are informative and very well organized.
The few remarks that in my opinion can improve the manuscript are enlisted below:
Line 221 – remove the comma before the sentence.
Line 222 – the authors have to indicate about the number of the logarithm, for example |Log2FC|>1
Line 246 – there should not be a space between “viable” and the dot.
Line 246 – In the sentence “G-SCs showed the highest proliferation ability, followed by P-SCs (Figure 1C).” However, it is showed in contrary on the figure or P-SCs proliferates faster than G-SCs.
Line 282 - The word "slightly" should be removed, as there is an obvious difference between the OD of HSC-3+G-SCs and HSC-3+P-SCs.
Line 445 – There are two commas after “Second”.
Line 452 – “Finally, s HDFs” should become “Finally, HDFs”
In my opinion, the current version of the manuscript is suitable for publishing in after minor corrections.
Author Response
Reviewer2
The manuscript presented by Omori et al, entitled “ The Origin of Stroma Influences the Biological Characteristics of Oral Squamous Cell Carcinoma” describes the role of stromal cells derived from the gingival and periodontal ligament tissues in the proliferation, invasion, and migration of OSCC both in vitro and in vivo.
The authors present quite intricate data, that demonstrate for the first time that the normal stroma (G-SCs and P-SCs) is directly associated with changes in the biological characteristics of OSCC tumor parenchyma, such as proliferation, invasion, and morphology, both in vitro and in vivo. All this has the potential to be applied in the clinical practice as therapeutic strategy for OSCC.
Overall, their results indicated that both G-SCs and P-SCs could promote the progression of OSCC, which could be a potential regulatory mechanism in the progression of this malignancy.
In general, the material is well organized and easy to be followed and understood.
The results are clear and satisfactory.
The figure and the graphics are informative and very well organized.
The few remarks that in my opinion can improve the manuscript are enlisted below:
Line 221 – remove the comma before the sentence.
Response: The comma has been removed, please check it.
Line 222 – the authors have to indicate about the number of the logarithm, for example |Log2FC|>1
Response: The |LogFC| has been changed into |Log2FC|, please check it.
Line 246 – there should not be a space between “viable” and the dot.
Response: The space has been deleted, please check it.
Line 246 – In the sentence “G-SCs showed the highest proliferation ability, followed by P-SCs (Figure 1C).” However, it is showed in contrary on the figure or P-SCs proliferates faster than G-SCs.
Response: The wrong description for the proliferation ability of G-SCs and P-SCs has been modified, please check it.
Line 282 - The word "slightly" should be removed, as there is an obvious difference between the OD of HSC-3+G-SCs and HSC-3+P-SCs.
Response: The word slightly has been removed, please check it.
Line 445 – There are two commas after “Second”.
Response: The redundant comma has been removed, please check it.
Line 452 – “Finally, s HDFs” should become “Finally, HDFs”
Response: Finally, s HDFs” has been changed into “Finally, HDFs”, please check it.
In my opinion, the current version of the manuscript is suitable for publishing in after minor corrections.
